# Hospital Ownership and Hospital Institutional Change: A Qualitative Study in Guizhou Province, China

**DOI:** 10.3390/ijerph16081460

**Published:** 2019-04-24

**Authors:** Yu Xie, Di Liang, Jiayan Huang, Jiajie Jin

**Affiliations:** School of Public Health, Fudan University/Key Lab of Health Technology Assessment, National Health Commission, Shanghai 200032, China; pumc1917@126.com (Y.X.); 15211020032@fudan.edu.cn (J.J.); Department of Family Medicine and Public Health, University of California, San Diego, CA 92093, USA; dil067@ucsd.edu

**Keywords:** property right, ownership, organizational reform, case study, Guizhou, China

## Abstract

*Objectives*: To qualitatively compare the influence of different ownership which is considered as a kind of institutional environment in public hospitals, private hospitals, and mixed-ownership hospitals on hospital governance structure and organizational behavior. *Design*: Qualitative descriptive study, using semi-structured, in-depth interviews and thematic template analysis, theoretically informed by critical realism. *Participants*: 27 key informants including national policymakers in charge of the health sector, influential researchers, local administrators responsible for implementing policies, and hospital managers who are experienced in institutional change. *Results*: Hospital ownership has a significant influence on hospitals in terms of decision-making power allocation, residual ownership allocation, market entry level, accountability, and social functions. These five aspects in hospital organizational structure incentivize hospitals to adapt to the internal and external environment of the hospital organization—such as market environment, governance, and financing arrangements—affect the behavior of the hospital organization, and ultimately affect the efficiency of hospital operation and quality of service. The incentives under the public system are relatively distorted. Private hospitals have poor performance in failing their social functions due to their insufficient development ability. Compared to them, mixed ownership hospitals have a better performance in terms of incentive mechanism and organizational development. *Conclusion*: Public hospitals should improve the governance environment and decision-making structure, so as to balance their implementation of social functions and achieve favorable organizational development. For private hospitals, in addition to the optimization of the policy environment, attempts should be made to strengthen their supervision. The development of mixed-ownership hospitals should be oriented towards socialized governance.

## 1. Introduction

The public sector in industrialized countries (during the 1960s and 1970s) and in developing countries (during the 1980s) went through a series of transformations [1,2,3,4,5], against the backdrop of the reform of ‘new public management’ or ‘marketization’. These transformations included increasing the management autonomy of the organization, transforming the hierarchical bureaucracy into parastatal corporations that are exposed to competition pressures, and complete detachment from the public sector. Public hospitals also experienced autonomization, and many became corporations, aiming at efficiency, equity, and quality improvements. However, such transformations do not change the public ownership of these hospitals. A systematic appraisal on these reforms has been performed by Preker [6].

China initiated public hospital reform under the influence of the same ideological trend [7]. In China, hospitals get classified for their capacity of medical care, medical education, and medical research with levels I through III [8]. Level I hospitals are also known as primary health service institutions which provide basic medical services, preventive services, and so on. Most level III hospitals are considered regional medical centers in charge of complex medical services. Level II hospitals are in-betweens. Patients with serious diseases can be transferred from level I to level II or even level III hospitals. Hospitals are also classified as public or private according to their ownership. Hospitals are also classified as public or private according to their ownership. Since the 1980s, Chinese public hospitals, including hospitals affiliated with state-owned enterprises, have experienced autonomization or privatization [9,10]. In the past decade, Chinese government introspected the previous radical reform and were concerned about the political risks associated with the privatization of public hospitals. Therefore, public hospitals were encouraged to improve their efficiency and quality of care while the government did not lose the control of the property rights. Meanwhile, more private hospitals were encouraged to develop so as to create a ‘catfish effect’ by increasing competitiveness. 

Currently, some private investors are allowed to invest in public hospitals, making these hospitals under mixed ownership [11]. Mixed ownership hospitals is controversial. The government may suffer reputational damage from profit-seeking behavior of mixed ownership hospitals. However, most mixed ownership hospitals are essentially defined as non-profit hospitals in order to get preferential policies and tax deductions. Mixed ownership hospitals are like joint-stock companies as they are jointly owned by the government and private investors. Thus, mixed ownership hospitals differ from traditional public hospitals and private hospitals, but have similarities in many respects. Because of the participation of private investors, mixed ownership hospitals have more autonomy and market behaviors than public hospitals. Consequently, mixed-ownership hospitals have the characteristics of autonomous units and privatized units in the organizational change framework defined by Alexander S. Preker and April Harding from the World Bank.

The organizational reform framework was a rich synthesis of neoclassical economics, principal-agent theory, transaction cost economics, property rights theory, and public choice theory. This framework could be used as the basic framework for evaluating public hospital reforms and is particularly suitable for analyzing property rights reform and governance structure or incentive changes in the process of public hospital reform. The framework includes decision-making power allocation, residual ownership allocation, market entry level, accountability, and social functions. As an important basis for other powers and activities, decision-making power includes the power to decide strategic objectives related to major hospital developments, financial investment, and management decisions (e.g., human resources, clinical and non-clinical business scope, market strategies, sales, and production) [6]. The residual claim refers to the right to claim the remaining income when all legal obligations (e.g., debts) have been fulfilled. The attribution of residual claim rights and the distribution of residual income constitute the most powerful incentive mechanism for owners. For instance, a hospital may extend its scale if this is the best way to increase its revenue. Decision rights and residual claims differ in hospitals with different ownerships and determine hospitals’ governance structures and incentive mechanisms. Market entry level refers to earning revenue under market conditions instead of relying only on budget allocation. Market exposure is the environment facing the hospital which also decides hospitals’ ability to adapt to marketization. Accountability can be executed through hierarchical supervision of the organization, regulations, and other indirect mechanisms such as contracting and procurement. Accountability mechanisms are mainly used to regulate or guide hospital behaviors. For instance, fixed investment from the government is an important regulation tool for the government in case of market failures to encourage public hospitals to provide certain services through cross-subsidization (similar to some quasi-public products in some cases) [12]. These services are often less cost-effective but have better social effects, such as emergency services, training of medical talents for society, and free treatment for the poor. Hospitals with different ownerships, therefore, fulfil their social functions to different extents. 

Internationally, previous studies mainly focused on comparing for-profit and non-profit medical institutions. A systematic review and meta-analysis suggested that since 1990, hospitals of different ownerships in the United States have not shown significant differences in financial performance, including cost, revenue, profit margin, and efficiency [13]. Similarly, another systematic review found that private non-profit hospitals might be slightly better than public hospitals and private for-profit hospitals in the United States in terms of quality of care measured by mortality and patients’ adverse reactions [14]. A possible explanation might be that hospitals of different ownership behave differently in response to different financial incentives [15]. In China, many studies showed that public and private for-profit medical institutions had no significant differences in medical cost [16] and resident satisfaction [17,18]. However, most previous studies, especially those quantitative ones, did not examine how ownership affected financial performance and quality of care through contextual circumstances and operations [19]. Furthermore, there was no previous research examining the consequences of mixed ownership for hospitals. 

To fill the knowledge gap above, this study considers the ownership of a hospital as a kind of institutional environment and analyzes the influence of different ownerships on hospital governance structure and organizational behavior. Hospital governance structure, including decision-making power allocation and residual ownership allocation, is interrelated to organizational behaviors, such as market exposure, accountability, and social functions. In order to better compare across ownerships and illustrate the features of mixed ownership hospitals, public hospitals and private hospitals are selected as comparison cases. Guizhou Province was chosen as a case for the following reasons. First, Guizhou Province is a typical example of the areas lacking medical resources, and these areas better represent the developmental stage of China. Second, Guizhou Province also has all three categories of hospital ownership investigated by this study. Furthermore, although such cases also exist in other provinces in China, the established cooperation between Guizhou and our research group leads to the choice. 

This study is one of the cases from the research project titled “Research on the Effectiveness and Strategy of Government Regulations for Private Healthcare in the Background of New Medical Reform”, which is funded by the National Natural Science Foundation of China.

## 2. Methods

### 2.1. Data Collection 

Data was collected for this research from peer-reviewed literature, secondary statistical data, policy documents related to national and local hospital regulation used in policy text mining, and face-to-face in-depth interviews. Based on literature review and the research objectives, a set of questions were developed for key informants. We used stakeholder analysis and snowball sampling methods to identify informants who adequately represented the key stakeholders [20]. The key informants selected and interviewed cover policymakers at the national level, municipal administrators, hospital managers, and researchers on hospital property reform. Table 1 describes their details. Administrators from Guizhou Province clearly knew the local conditions because they directed hospital reforms and implemented routine regulation on local hospitals including private hospitals, public hospitals, and mixed ownership hospitals. Policymakers from the Ministry of Health knew about hospital reforms on a nationwide scale that could provide information outside of Guizhou Province. Hospital managers came from hospitals both inside and outside Guizhou Province, and all of them were experienced in hospital reforms. 

Semi-structured in-depth interviews were conducted with 27 key stakeholders (Table 1) face-to-face (except one via telephone), including policymakers at the national level from hospital management and healthcare sectors, key researchers influencing policymaking, local administrators executing policy, and hospital managers who experience institution change most. The interviews took place at their workplaces and were conducted by two of the authors of this paper. Informants were not compensated for their time. Oral consent was also obtained for participating in the interview as well as audio recording. Each interview lasted for about 60 min and was recorded with permission apart from one which was not recorded but typewritten at the respondent’s request.

The topic guidelines for the interviews included questions on the participant’s experience and reflections on hospital reforms, status quo, problems, and suggestions for various types of hospitals (Appendix A). Using probes and follow-up questions, attention was directed to factors mentioned by the interviewees explaining the reasons for hospital institution change.

### 2.2. Data Analysis

This study adopts a conceptual framework of reference consisting of a set of five organizational reforms of hospitals, which was developed by April Harding and Alexander S. Preker [6]. The framework has been used for evaluating various types of hospitals, including budgetary units, autonomous units, corporatized units, and privatized units. The mixed ownership hospitals mentioned by the participants have the characteristics of both traditional public hospitals and private hospitals, and the five principles are suitable for this study. The principles are as follows: decision rights, market exposure, residual claims, accountability, and social functions. The answers from these key aspects were drawn from the interviews.

Audiotapes of the interviews were transcribed and compared with the field notes to check their accuracy. The analysis of data from the interviews included the following steps [21]: (1) be familiar with the data and understand the meaning of what was said; (2) identify emerging topics; (3) develop a topic index; (4) use the index to code the data; (5) consolidate the topics into themes; (6) further consolidate the themes into analytical categories; and (7) translate the analysis obtained into a narrative. 

This research used MindManager to manage qualitative data and analyze the themes of perceptions using thematic template analysis, which is a method to identify and report patterns in qualitative data collected from different interviewees [22]. QSR NVivo 10.0 (QSR International Pty Ltd, Melbourne, Australia) was used to assist the analysis and facilitate the coding process. The raw data were coded by two independent reviewers (Y.X. and J.J.). If divergence emerged, a third reviewer (J.H.) would participate in the group discussion until the group reached a consensus. This research analyzed governance structure and organization behavior according to the theoretical framework of the five key principles of hospital organization reform.

Data collection was restricted to approved members of the research team who signed a confidential agreement with the principal investigator. Data were securely stored in electronic form. Data processing was kept anonymous so as to protect the privacy of interviewees. The names of the respondents were deleted from the quotations.

### 2.3. Patient and Public Involvement

There was no patient or public involvement in this research project.

## 3. Results

Since the reform and opening up, the environment for medical institutions in China has undergone intense and frequent changes, and hospitals have been more exposed to competition^20^. The Chinese government still adheres to the traditional centralized system, yet also appropriately decentralizes to prompt hospital reform. (1) Some public hospitals began the process of privatization. In particular, many hospitals affiliated with state-owned enterprises changed their property during the reform of state-owned enterprises [23]. (2) While maintaining their property rights [24], some public hospitals cooperated with private investors to build public–private partnerships, to different extents or in different forms. (3) In some hospitals, the original status has been maintained [25], but significant changes have taken place in their behaviors. All these types of hospitals share a common goal of accommodating to the special institutional environment of China, and the changes they have made for this purpose are shown in detail below (Table 2).

### 3.1. Decision Rights

Public hospitals in China have a deep symbiosis with the government. Public hospitals obtain resources from the government, such as land allocations, talent recruitment, financial investment, and support for disciplinary development [26]. Admittedly, this dependency is at the expense of decision-making authority. Major decisions—such as infrastructure construction and hospital development direction, the recruitment and dismissal of hospital personnel, service delivery content, etc.—are subject to the governmental intervention at different levels and of different types. In the early stage of China’s new health system reform, it was determined that power decentralization was an important part of the reform of public hospitals. However, the power decentralization did not change the dependency of public hospitals on the government and the public institution system, producing very few effects [27].

Cooperation with private investors or direct privatization reforms have significantly changed previous ownership attributes, thus reshaping the ownership of decision-making authority. Although the reform of the former kind still maintains a non-profit nature and public hospitals still play the dominant role, the channels for new mixed ownership hospitals to obtain resources become diversified, just like companies introducing strategic investors. With new participants, the original governmental departments cannot intervene with new mixed ownership hospitals without permission. Hospitals after full privatization fall into the category of private hospitals, and the ways of accessing resources has completely changed. The change in decision-making authority depends on the degree of power possessed by new investors. In this study, Hospital E is a completely privatized hospital, and its investors hold the concurrent post as the managers of the hospital so that they have comprehensive decision-making authority over the macro-development and daily management of the hospital. The decision-making authority of new mixed ownership non-profit hospitals has increased tremendously: significant improvements have been achieved in terms of infrastructure construction, equipment purchase, independent pricing or recruitment, and service content, etc., and the improvements have also brought about a significant growth in services

Hospital B which was co-invested by university affiliated hospitals and a pharmaceutical enterprise receives talent support from the headquarters. The managers of Hospital B confirmed that “doctors in new hospitals sign work contracts with medical school, but their work and salary etc. are arranged by us”.

In this case, Hospital C was restructured from a staff-worker hospital affiliated to a state-owned enterprise through the investment and registration of an enterprise in Guizhou to form a new non-profit hospital. Under the new property rights structure, the rigorous control of government on mixed ownership hospitals has greatly eased because of the addition of new participants, such as enterprises. After the restructuring was completed and the new hospital was formed, many miracles happened in Hospital C: the renovation of old wards, the expansion of the inpatient building (232 new beds), the renovation of the administrative building (80 new beds), the thematic project of the hospital’s inpatient multiple-use building (600 new beds), and many new projects. Such expansion is unimaginable in a public hospital, because the government imposes strict control on the expansion of public hospitals, and even construction funded by the debt taken out by the public hospitals themselves is not allowed [28]. Although this policy of restricting expansion is necessary for economically developed areas with rich medical resources, it can be a constraint for some poverty-stricken areas. Therefore, the change in ownership has brought about a major change in decision-making authority, which is necessary for hospitals in remote areas to expand their scale and increase service supply, thus considerably lifting the ‘one-size-fits-all’ policy restriction. In addition, public hospitals must participate in centralized bidding, but new hospitals can engage in independent purchasing and gain the profits from the purchase.

One head of Hospital B also said, “I think the most prominent advantage of this favorable policy is that the government has no control on us and pays no special attention to us, so we can grow up with our strength”.

### 3.2. Residual Claim

For public and non-profit hospitals, the law that the balance of revenues cannot be distributed means no residual claim. Therefore, despite an increase in the autonomy of decision-making, the lack of residual claim greatly diminishes the incentive effect of additional autonomy. Nevertheless, in public hospitals, many incentives are obtained indirectly through residual control rights, such as on-the-job consumption and pay raises for senior executives or employees in spite of the lack of residual claims. Of course, this behavior is under strict control [29]. Therefore, public hospitals lack the motivation to save costs, and they attempt to use their remaining control rights to expand the scale of public hospitals and control more resources, forming a vicious circle [30].

Similar to public hospitals, although legally unable to obtain claims for balance of income and expenditure, private non-profit hospitals which have gone through property rights reforms have control over their hospitals. They may obtain profits through exclusive access to services such as drug supply chains, hospital infrastructure construction, etc., or indirectly gain profits through the capital market or stock market. Such a practice not only represents the pursuit for the incentives of the remaining control rights, but also evades legal regulations and restrictions on public hospitals.

A president of mixed ownership hospital E said frankly, “private non-profit hospitals make profits mainly through the price differences. Take the purchase of a set of equipment as an example, a public hospital makes account at the price of 1000 RMB, but the actual cost is 300 RMB, so the revenue for the hospital is 700 RMB.”

A staff member in middle management of mixed ownership Hospital B suggested that “listed companies are still preferred, because they do not need dividends and all they care are numbers and performance. Capital investment is profit-oriented, but the profits may not come directly from the hospital and can be sought from other areas of the market.”

Admittedly, for-profit hospitals have residual claims, but they have insufficient revenue derived from patients due to the lack of top-quality medical personnel. In terms of expenditure, they need to pay income tax, and they are disadvantageous in terms of land utilization (covering a large area) and loaning, financing, and mortgage. Therefore, their balance of income and expenditure is limited.

### 3.3. Market Entry Level

In the market economy system, hospitals of whatever ownership must face a competitive environment in which revenue is obtained under market conditions. In terms of revenue, public hospitals may rely on medical service charges as well as government budget and funding allocation, while their expenditure is affected more by the market environment [31]. Of course, the pressure on income and expenditure is very limited under the mechanism of soft budget constraints. With the addition of private investors, public hospitals have gradually shifted from budgetary values to autonomization or the form of legal personhood. Although some hospitals are still dependent on the government for talent introduction or staff rosters, they are apparently more autonomous and independent financially. For example, their fund raising is no longer subject to regulations concerning the fund raising for public hospitals, and the budget shifts from soft constraints to hard constraints. In addition, they need to face various pressures, such as market competition, independently.

As cooperate with private investors, Hospital B obtain the funds for infrastructure construction and equipment renewal. On the one hand, they have a direct access to the investment of private investors. For example, after the introduction of private investors by the affiliated hospital of the medical school, it received a fund of about 200 million RMB for the hospital’s infrastructure construction and equipment renewal. On the other hand, they can make use of private investors for fund raising, such as loans.

Hospital B’s managers confirmed that “our hospital cannot survive without the enterprise’s financial support and secured loans” and “in terms of capital, we mainly rely on private investors to maintain operation, but from another perspective, we are still a directly affiliated hospital. In terms of management, we still get some good treatment from public hospitals, which is sort of transfer payment. Although there was no direct funding at that time, policy support was given: at least the name of Guizhou Cancer Hospital was granted.”

Of course, the new mixed ownership hospital needs to independently face financial pressure, and it can increase its revenue by improving services, attracting patients, and improving the patient turnover so as to provide favorable conditions for better residual control and claims.

Hospital B “maintains an annual revenue growth of 15%, and strives to reach an economic aggregate of 500 million RMB in five years”. “From January 2016 to November 2016, the total revenue was 625 million RMB, representing a year-on-year increase of 16.1%.” The number of beds in Hospital A increased from 248 in 2010 to 649 in 2015. The number of people utilizing the outpatient service increased from 70,000 in 2010 to 283,668 in 2015, and the number of turnovers (patients admitted to and discharged from hospital) increased from 5148 to 34,916.

In terms of expenditure, “our control on management cost is more accurate than that in some public hospitals, so the use of governmental health insurance funds is actually more efficient.” (Manager-2 of Hospital B)

It should be admitted that although the introduction of private investors in public hospitals has increased their independence under the market conditions, they still enjoy their previous advantages of resources such as land and talents inherent in public hospitals, and intangible assets brought by reputation of public hospitals, still giving them an advantageous position in market competition.

Private hospitals have a higher degree of market entry and integrate medical resources entirely through capital. However, its disadvantages and pressure are also very apparent. First of all, in terms of revenue, since factors, such as difficulty in obtaining medical insurance designated qualification and lack of talents, limit the ability to attract patients.

“It can be very difficult to operate in the first one or two years because new private hospitals cannot attract patients in a short time and many service items are not covered by medical insurance, so they are always in debt.” (Local Health Administrator-1)

“The main source of patients is mainly from introduction by acquaintance, some patients are covered by medical insurance, but most of them are patients at their own expense (half approximately) and usually pay the visit to see a specific expert in our hospital.” (Manager-1 of Hospital E)

Of course, private hospitals can also make profits using the above-mentioned methods adopted by the non-profit hospitals, but their ability is limited by the above conditions. Therefore, the source of patients is very important for the revenue structure of private hospitals.

“The real problem does not lie in the lack of talents, but in the lack of patients. As long as the source of patients can be broadened and money is pouring in, we can hire high-end talents with high salaries.” “However, in some cases, in order to solve this problem, some hospitals have taken illegal measures: we earlier heard from some employees that some private hospitals arranged hospital scalpers at famous hospitals and then directly picked up some patients with their car.” (Local Health Administrative-1)

Also, private hospitals are more stringent in cost control. Even “public hospitals make it compulsory to buy malpractice medical insurance, it is still voluntary in some private hospitals” (Manager-2 of Hospital E). In conclusion, the pressure of private hospitals in the face of the market is considerable, and the Manager-1 of Hospital E believes that “if the policy is favorable, we can turn loss into profit in half a year, but based on the current situation, the time will be at least two years”.

### 3.4. Accountability

There are obvious differences and similarities in the accountability of various types of hospitals. In terms of differences, as the government has comprehensive control over the decision-making authority of public hospitals, the government’s accountability for public hospitals is achieved through bureaucratic or administrative direct accountability, such as direct appointment of hospital leadership, which is a “patriarchal” accountability approach in China’s institutional environment [32]. As for private hospitals, which are less government-dependent, the government is more likely to adopt indirect mechanisms, such as economic methods, the mechanism of regulatory accountability, purchasing services and so on. For hospitals with mixed ownership, their original paternal accountability is made less apparent due to the addition of new participants or changes in the subordinate departments, and other ways of accountability become strengthened. There is also another important accountability method, which is an interdependent social network relationship formed during the long-term attachment in both bureaucratic management and administrative management [33]. The trust mechanism also represents a constraint on both parties. It still exists in traditional public hospitals and mixed-ownership hospitals that still retain public shares, while private hospitals need to construct new relationship models.

The similarity is manifested in two aspects: firstly, the accountability of medical insurance regulation plays an increasingly important role. Whichever type the institution falls into, the designated access and payment methods of medical insurance have been increasingly served as methods to supervise medical behaviors, resulting in more obvious effects. The reason behind this phenomenon does not lie in the change of the nature of the organization, but in the increasing influence of medical insurance on medical institutions [34]. Secondly, in China, both public and private hospitals are subject to the same industrial regulation.

The local health administration Manager-2 confirmed that “our supervision on private hospitals is the same as that on public hospitals. The supervision is very strict. If there is a problem that needs to be rectified, it must be rectified.... So private hospitals and public hospitals are treated in the same way in recent years. The requirements for both public hospitals and private hospitals are the same.”

However, public hospitals and mixed ownership hospitals in fact enjoy certain tolerance because of “paternalistic” supervision, while private hospitals are inadequately supervised due to weak regulatory efforts and poor capacity [35], thus causing some tragic incidents in illegal private hospitals, such as the Wei Zexi tragedy [12].

The Local Health Administration Manager-2 in charge of the health supervision department said frankly: “To our knowledge, we feel that the government supervision has not been fully implemented...; for purely private hospitals, only spot checks are feasible. At present, the government’s supervision of purely private hospitals is very difficult, and supervision is not necessarily complete, because these hospitals cooperate poorly with the government, and penalties are not likely to be imposed to them, so the measure adopted each time is mutual examination among different hospitals.”

Accountability is an important guarantee for the order of the medical service market and the quality of medical services. In fact, no matter which category a hospital belongs to, the indirect accountability approaches—such as reducing the accountability mechanism of direct administrative intervention and adopting the rules that regulate the market of medical services according to market development—are critical to the healthy and sound development of China’s health care system.

### 3.5. Social Function

During the process of becoming a mixed ownership hospital, the government’s investment has obviously reduced or disappeared, which is followed by an increase in the degree of marketization, so social functions are very likely to diminish [36]. However, based on the consideration of interests and the need to take up the market share, hospitals will still provide some social function services that can obviously affect residents and government senses out of the great concern about market and competition, and because of information asymmetry, some less noticeable details are omitted and services with a greater impact on the operation of the organization are not provided. In this case, mixed-mode hospitals behaved in a friendlier manner than the previous public hospitals in terms of attitudes toward patients, because under the conditions of hard budget constraints and market competition, striving for patients is one of their fundamental measures, and this consideration objectively increases patients’ welfare. Besides, Hospital B, Hospital C, and Hospital D still undertake some social welfare programs by providing the services which are originally provided by the public hospitals, and it may adopt some marketing methods.

“These measures include free medical assistance to the citizens with no income, no labor capacity and no legal obligations, and medical treatment of patients in exceptional poverty, medicine delivered to the rural areas and free clinic activities, and pairing with the TCM hospitals in the Bijie county to support them.” (Manager-2 of Hospital C)

Private hospitals do not have such considerations and are completely market-oriented. They focus on problems like how to avoid the need to conduct standardized training for residents to reduce costs and avoid the risk of training doctors for other hospitals.

“Now it is too difficult for us to introduce talents. We can only let them first attend the training programs and hire them later by offering a high salary.” (Manager-3 of Hospital E)

## 4. Discussion

This study analyzes the materials collected from the interviews with major domestic stakeholders and the changes in organizational structure and behavior of hospitals of different ownerships under the framework of hospital organizational reform. It shows that hospital ownership has significant effects on decision-making authority, residual claims, market entry level, accountability, and social functions, and the five aspects in hospital organizational structure represent the key incentives in hospital reform, which together with the external environment faced by hospital organizations—such as market environment, governance, and financing arrangements—affect the behavior of hospital organizations and ultimately may influence the efficiency and quality of service. Consequently, the behavior of different ownership hospitals varies.

(1) Although public hospitals have a dependency relationship with the government which provides easy access to abundant policies, talents, and financial resources that they need for survival and development, their organizational structure and behavior are greatly restricted by the government, which is demonstrated by the government’s intervention with the decision-making authority of public hospitals, the prohibition of residual claims, and the indirect restriction of its residual control right by limiting the scale expansion. In this institutional environment, public hospitals have neither the motivation nor the ability to participate in market competition. Their main competition means is lobbying for resources from the government [27]. Of course, the government can intervene with the public hospitals through the “one-vote veto” administrative intervention as well as the long-established trust mechanism of social network [37]. The exercise of social functions is an inertial behavior.

(2) The governance of private hospitals is a relatively simple process, but in the face of a market environment without stable financial investment from the government, their survival depends on the decision of investors. As a result, the incentive for residual claims is intense, but this incentive is greatly restricted by the large amount of new fixed-cost inputs, limited market share, and unstable sources of funding. In order to achieve such incentive, private hospitals, on the one hand, cut costs and avoid risks, on the other hand, they strive to selectively provide competitive services, which greatly compromises their performance of social functions.

(3) Mixed ownership hospitals, although still subject to government intervention, have greatly increased autonomy. In particular, the administrative intervention under the current management system on personnel compensation, investment and financing, pharmaceuticals, infrastructure construction, and equipment procurement in public hospitals is less evident in mixed-ownership hospitals. Without residual claims, they can still carry out construction or obtain indirect income from capital markets through the drug supply chain and hospital infrastructure construction (although some behavior is more of a gray area), and at the same time they retain the talents, medical insurance, financing, and some policy support inherit from the original public hospitals. Their external environment and internal environment provide sufficient boost for their development. Although the paternalistic accountability system still exists, they can exchange information with the government smoothly due to the trust mechanism of social networks, and at the same time the hospitals can also undertake some social functions.

In summary, hospitals of different ownerships adapt to changes in the external environment of the organization through organizational structure and behavior reforms, ensuring that the organization can survive and develop. Compared with other studies focusing on hospital ownership with regard to research and efficiency [38], the comparison of quality [39], or the analysis of the performance of some social functions [40], this study mainly describes the organizational structural adjustments and behavioral changes of hospitals of different ownership structures in the face of the environment of property rights. It does not conclude a direct relationship between structural change and performance, but points to the relationship between market environment changes and incentive directions. Recent studies focus on the association between hospital ownership and the efficiency and quality of hospitals, which show how hospitals of different ownership behave strategically in response to the financial incentives provided by the health care payment policies. Our study finds that hospitals of different ownership in China are treated differently in terms of health insurance policies, which may lead to different financial incentives [41]. In addition, we describe the mixed-ownership hospitals as a unique transitional form of medical organizations in countries going through transition, enriching Preker’s patterns of hospital organizational reform [6].

### 4.1. Reform Direction of Chinese Medical Organizations

Different from general classification of hospitals as public hospitals and private hospitals which can be subdivided as private for-profit hospitals and private non-profit hospitals, Chinese hospital organizations are categorized into public hospitals, private hospitals (for-profit or non-profit), and mixed-ownership hospitals. Mixed-ownership hospitals should be a transitional form of hospital organization as China undergoes economic and social transformation. At present, hospitals that receive investment from private investors are still profit-oriented. The private investors either directly obtain profits through residual claims in for-profit hospitals, or earn profits from supply chain links such as drugs, consumables, and medical equipment through residual control rights in non-profit hospitals, or gain large profits by using their appeal in the market in public hospitals and public hospitals. All these measures are legally ambiguous and not subject to the supervision and accountability. The purpose of government regulation is not to prevent hospitals from making profits, but to limit them to the scope of legal compliance. At the same time, it can guide medical institutions to perform some social functions through accountability measures, such as purchasing services and issuing favorable policies. Therefore, supervision and accountability should be strengthened if choosing mixed ownership.

Because public governance has been introduced to China for only a short period of time and there is a severe lack of the participation of residents, the cooperation between government and external participants to run non-profit hospitals can be a good way to initiate the reform of corporate governance. The current non-profit hospitals in the current joint-stock system will implement the corporate governance structure of public corporations in Japan [42], and the shareholding system can be used as the weight of each participant’s control over the hospital, but it is still a transitional form. It is possible to explore the gradual transition to the participation and influence of legal representatives according to shares, which can be reflected in the right of setting the number of directors or members, and the transformation of joint-stock non-profit hospitals into non-profit hospitals of public welfare corporations. At the same time, the government should formulate incentive policies such as tax reduction and exemption for companies or individuals who donate to or ‘invest in’ non-profit hospitals.

### 4.2. Research Limitations

The limitations of this study are mainly reflected in two aspects: Firstly, in this study, we only selected a provincial case in western China, though there are more mixed ownership hospitals in rural areas where more public hospitals operate poorly and lack inputs from the government and private capital was encouraged to join in. Meanwhile, Appendix A such as the interviews of national regulators, managers of grade A class 3 hospitals and some experts, were used to interpret more reforms beyond the province. Geographical difference and unbalanced economic development in China also weakened the explanatory power of the conclusion. Secondly, due to the requirements for confidentiality on some information of the data and interviewees, the issue is not fully discussed in this study. Further efforts are needed to carry out in-depth research into the relationship between hospital ownership and organizational reform.

## 5. Conclusions

This study explores the relationship between hospital ownership and hospital institutional change and expands the research on hospital property rights to the relationship between the environment of the organization and the structure and organizational behavior of the hospital.

Our study explores institutional reform in hospitals under different types of ownership in China and the views of key stakeholders concerning those types of ownership and their influence on hospitals’ behavior. These findings are essential in choosing the right pathway for Chinese medical organization reform.

Our study provides a new dimension for understanding hospital reform in China, a country undergoing transition.

Still, the selection of participants may introduce some bias to our studies, and some efforts are needed to study the depth of the relationship between hospital ownership and organizational reform.

## Figures and Tables

**Table 1 ijerph-16-01460-t001:** Number of interviewees listed in order of administrative level and facility.

Types of Interviewees	Level	Number of Participants
Policymakers		
Ministry of Health	National	3
Researchers	Two universities	2
Administrators	Guizhou Province	2
Hospital managers		
General hospital A	Level III, national, public	3
General hospital B	Level III, provincial, mixed ownership	5
General hospital C	Level II, provincial, mixed ownership	3
Specialized hospital D	Level III, provincial, mixed ownership	3
Private hospital E	Level II, provincial, private	6
Total		27

**Table 2 ijerph-16-01460-t002:** Comparison of the organizational reform in hospitals of different categories

Frame	Public Hospitals	Private Hospitals	Mixed Ownership Hospitals
Decision-making authority	Public hospitals’ decisions on their major development and daily management are often subject to government intervention	Hospital investors and managers have full decision-making authority over the major development and daily management of the hospitals	Governmental intervention with hospital management is greatly reduced, and mixed-ownership hospitals have greater autonomy in major development and daily management decisions
Residual claim	They do not have residual claim but possess residual control right and feature a high amount of waste as well as the stimulus to expand	They have residual claims, but the limitation on operating conditions results in limited balance	They do not have residual claim, but they carry out construction or obtain indirect income from the capital market through the drug supply chain and hospital infrastructure construction
Market entry level	In a dual environment where planned economy and market economy coexist, the market plays a dominant role in some aspects, while in other aspects its role can be weak	These hospitals have the highest market entry level and can integrate resources through capital, but factors such as difficulty in obtaining medical insurance designated qualification and lack of talent limit their competitiveness	Such hospitals not only inherit the talents, medical insurance advantages, and some government support from the original public hospitals, but also gain market flexibility under the new system, so they are relatively more competitive in the market
Accountability	Approaches include direct administrative accountability, strong patriarchy, involving supervision and medical insurance payments as well as the trust mechanisms of social network	Approaches include economic accountability, supervision, and medical insurance payments	Economic accountability, supervision, and medical insurance payments are the mainstays, and the role of paternalistic accountability is intermittent; trust mechanisms of social networks still exist
Social function	They undertake the government’s medical security responsibilities and provide less cost-effective but socially beneficial services through cross-subsidization	They focus on profits and provide services that help to enhance their competitiveness in order to reduce costs and avoid risks	They provide some social function services and focus on striving for patients

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
