# Peer review of "Hospital Ownership and Hospital Institutional Change: A Qualitative Study in Guizhou Province, China"

_ijerph, 2019, doi:10.3390/ijerph16081460_

Round 1

Reviewer 1 Report

In the present study the authors performed qualitative interviews with different stakeholder groups in order to explore the effect of hospital ownership (public vs. private vs. mixed) on organizational structures and processes of hospitals in Guizhou province of China.

While the topic is interesting, major revision is indicated before this manuscript can be considered for publication.

The manuscript would certainly benefit from a language review by a native speaker as the meaning of several passages in the manuscript remains unclear and grammatical errors abound.

Since Preker’s framework seems to play a large role in this study in terms of theoretical background as well as framework for the interview guide and analysis, it is suggested that this framework is described in the introduction.

In the introduction you write that previous research has focused on economic effects and quality of care and that future studies should focus on the relation of environment and performance. What do you mean by that? To analyze the relation of hospital ownership with quality of care seems to focus on a relation of environment and performance.

It also seems necessary to explain the Chinese hospital system in more detail in the background section (i.e. level III, II, I hospitals). Later you speak of tertiary and secondary hospitals but you should call it level III and II hospitals since it is otherwise easily confused with tertiary care hospitals etc.

The objective and specific aims of the study need to be more clearly outlined, also with respect to implications the study may have in terms of future policies.

Please better describe the interview guide and it’s development and provide it in the Appendix. What do you understand by policy text mining?

You write that key informants included researchers but I can not find any in Table 1.

Better describe the analysis, was it inductive or deductive or mixed. It seems you took Preker’s framework as a broad roster and then coded emerging topics within this roster if I am right. How was this done? I miss a systematic analysis and description of emerging topics in results.

The results need to be better structured, i.e. sub-topics within the Preker framework need to be systematically presented and grounded in the data, i.e. more quotations are needed to illustrate findings.

L 158 What do you mean by the Chinese government adhering to a traditional unit system?

It is strange to speak of social capital when you actually seem to mean private funds.

In line 274 you write that public hospitals rely on private capital. I do not understand.

The paragraph strength and limitations of the study seems rather a summary.

Author Response

Point 1: While the topic is interesting, major revision is indicated before this manuscript can be considered for publication.The manuscript would certainly benefit from a language review by a native speaker as the meaning of several passages in the manuscript remains unclear and grammatical errors abound.

Response 1: Thanks for your kind suggestion. We have checked the whole paper. If further language editing needed to be done, we will check again.

Point 2: Since Preker’s framework seems to play a large role in this study in terms of theoretical background as well as framework for the interview guide and analysis, it is suggested that this framework is described in the introduction.

Response 2: Thank you very much for pointing out important issues. We have added description about this framework in the introduction from line 90 to line 112, line 159, line 160.

Point 3: In the introduction you write that previous research has focused on economic effects and quality of care and that future studies should focus on the relation of environment and performance. What do you mean by that? To analyze the relation of hospital ownership with quality of care seems to focus on a relation of environment and performance.

Response 3: Thank you very much for pointing out important issues. We rectified the sentence in order to make the reader identify the aim and the process of the study as follows: this study considers ownership of hospitals as a kind of institutional environment, and analyze the influence of different ownership on hospital governance structure including decision-making power allocation, residual ownership allocation which shows different organizational behavior in market exposure, accountability and social functions.

Point 4:It also seems necessary to explain the Chinese hospital system in more detail in the background section (i.e. level III, II, I hospitals). Later you speak of tertiary and secondary hospitals but you should call it level III and II hospitals since it is otherwise easily confused with tertiary care hospitals etc.

Response 4: Thanks for your kind suggestion. We have added more details about the Chinese hospital system in the introduction section from line 42 to line 66, and used level III and II hospitals instead of tertiary and secondary hospitals.

Point 5:The objective and specific aims of the study need to be more clearly outlined, also with respect to implications the study may have in terms of future policies.

Response 5: Thank you very much for pointing out important issues. We rectified the sentence in order to make the reader identify the aim and the process of the study as follows: this study considers ownership of hospitals as a kind of institutional environment, and analyze the influence of different ownership on hospital governance structure including decision-making power allocation, residual ownership allocation which shows different organizational behavior in market exposure, accountability and social functions.

Point 6:Please better describe the interview guide and it’s development and provide it in the Appendix. What do you understand by policy text mining?

Response 6: Thank you for your kind suggestion. We added interview guide in the Appendix. We think policy text mining is one of data mining that could obtain valuable information from policy text. This methodology is very useful which could get quantitative result and qualitative result. Especially the keywords deeply mined from policy could uncover how the government want to execute the policy, and tell the inter logic of policy in Chinese institutional environment.

Point 7:You write that key informants included researchers but I can not find any in Table 1.

Response 7: Thank you very much for pointing out the issue. We corrected table 1 and outlined the researchers.

Point 8: Better describe the analysis, was it inductive or deductive or mixed. It seems you took Preker’s framework as a broad roster and then coded emerging topics within this roster if I am right. How was this done? I miss a systematic analysis and description of emerging topics in results.

Response 8: We used both inductive and deductive approaches in this study, that is mixed. We used the deductive approach by using Preker’s framework as a broad roster which concluded Chinese hospital organization reform. However, China is a transitional country where some is familiar with western country in general situation, and some is different from other countries. We used the inductive approach to extend Preker’s framework according to our data.

Point 9:The results need to be better structured, i.e. sub-topics within the Preker framework need to be systematically presented and grounded in the data, i.e. more quotations are needed to illustrate findings.

Response 9: Thanks for your kind suggestion. We compare three types of ownership under the framework in terms of decision-making power allocation, residual ownership allocation, market entry level, accountability, and social functions in the result section. It is difficulty to be further sub-division of the topics. We added more quotations to illustrate findings.

Point 10: L 158 What do you mean by the Chinese government adhering to a traditional unit system?

Response 10: Thanks for your kind suggestion. We have corrected the sentence and made it more clearly in line 346.

Point 11:It is strange to speak of social capital when you actually seem to mean private funds.

My response 11: Thank you very much for pointing out important issues. We used "private investors" instead of "social capital" to avoid confusing the reader.

Point 12: In line 274 you write that public hospitals rely on private capital. I do not understand.

Response 12: Thanks for your kind suggestion. We corrected the sentence as follows: “in terms of capital, we mainly rely on private investors to maintain operation, but from another perspective, we are still a directly affiliated hospital. ”

Point 13: The paragraph strength and limitations of the study seems rather a summary.

Response 13: Thank you very much for pointing out the issues. We added conclusion instead of strength and limitations.

Reviewer 2 Report

The purpose of the study is unclear, and the results are also insufficient. The number of hospitals involved in the analysis is too small and the number of particiants ratio based on the type of interviewee is not appropriate.

Author Response

Point 1: The purpose of the study is unclear, and the results are also insufficient.

Response 1: Thank you very much for pointing out important issues. We rectified the sentence in order to make the reader identify the aim and the process of the study as follows: this study considers ownership of hospitals as a kind of institutional environment, and analyze the influence of different ownership on hospital governance structure including decision-making power allocation, residual ownership allocation which shows different organizational behavior in market exposure, accountability and social functions.

 And we separated interviewees’ quotation from the other text in the result section.

Point 2: The number of hospitals involved in the analysis is too small and the number of particiants ratio based on the type of interviewee is not appropriate.

Response 2: Thank you very much for pointing out important issues. Refer to too small and the number of participants ratio, we added some interpretations why chose the interviewees in methodology section. We believe that the choose of interviewee is typical and representative. Although similar the inherent defects in all case studies that affected extending to other areas, the limitations would not affect the quality of the study.

Reviewer 3 Report

The paper addresses the issue of hospitals' governance and ownership within the Chinese healthcare sector, underlining its main strenghts and criticalities.

The research design is clear, but there are some parts in the paper that must be improved:

in the methodology section, a clear reference on the content analysis methodology must be done; 

in the discussion section, interviewees' quotations must be separated from the rest of the text. Moreover, sometimes, the subject who realeased the quotation is not indicated in parentheses, so the reader cannot correctly identify the interviewees' thoughts. A table resuming the different interviewees would be valuable so that the discussion could scroll better and the reader can better identify the point of view of different subjects that were involved in the study.

line 445: the following sentence is not clear and must be rephrased for some language errors: "The social capital either directly obtain profits through residual claims in for-profit  hospitals, or make profits from supply chain links such as drugs, consumables and medical  equipment through residual control rights in non-profit hospitals, or gain large profits by using their appeal in the market in public hospitals and public hospitals."  I suggest to change the term "social capital" within the paper with "investors", as "social capital" in healthcare literature can take very different meanings and its use in the paper risks to be confusing for the reader.

line 453: the following sentence must be cleared : "Therefore, the specific methods for mixed ownership hospitals need to be chosen with great cautions". What do you mean with "specific methods"? 

A professional language editing is required.

Author Response

Point 1: The paper addresses the issue of hospitals' governance and ownership within the Chinese healthcare sector, underlining its main strenghts and criticalities.

The research design is clear, but there are some parts in the paper that must be improved:

in the methodology section, a clear reference on the content analysis methodology must be done; 

Response 1: Thank you for your suggestions. We used thematic template analysis in the methodology section, not content analysis. We added description about thematic template analysis in line 293 to line 294 and reference literature numbered 22.

Point 2: in the discussion section, interviewees' quotations must be separated from the rest of the text. Moreover, sometimes, the subject who realeased the quotation is not indicated in parentheses, so the reader cannot correctly identify the interviewees' thoughts.

Response 2: Thanks for your kind suggestion. There are quotations in the result section, not in discussion section. We have separated the interviewees’ quotation from the rest of the text and formed separate paragraph in the result section.

Point 3: A table resuming the different interviewees would be valuable so that the discussion could scroll better and the reader can better identify the point of view of different subjects that were involved in the study.

Response 3: Thanks for your kind suggestion. We synthesized the views of 27 interviewees in the results section. The purpose of synthesis was to present the range of views across interviewees. Different interviewees might have different views but could also share many opinions. A table contrasting views of each interviewee might not be informative, as it might be much longer and messier than the current results section.

Point 4: line 445: the following sentence is not clear and must be rephrased for some language errors: "The social capital either directly obtain profits through residual claims in for-profit  hospitals, or make profits from supply chain links such as drugs, consumables and medical  equipment through residual control rights in non-profit hospitals, or gain large profits by using their appeal in the market in public hospitals and public hospitals."  I suggest to change the term "social capital" within the paper with "investors", as "social capital" in healthcare literature can take very different meanings and its use in the paper risks to be confusing for the reader.

Response 4: Thank you for your suggestions. We used "private investors" instead of "social capital" to avoid confusing the reader.

Point 5: line 453: the following sentence must be cleared : "Therefore, the specific methods for mixed ownership hospitals need to be chosen with great cautions". What do you mean with "specific methods"? 

Response 5: Thanks for your kind suggestion. We have corrected the sentence as follows: Therefore, supervision and accountability should be strengthened if choosing mixed ownership.

Point 6: A professional language editing is required.

Response 6: Thanks for your kind suggestion. We have checked the whole paper. If further language editing needed to be done, we will check again.

Round 2

Reviewer 2 Report

 The number of hospitals and interviewees(27 key stakeholders, Number of participants) involved in the analysis is too small. 

English sentences are not smooth.

For example, Line 45 to 48, English word duplication is awkward.

(Level I provides.. level III provides.., public hospitals and private hospitals, public hospitals can)

Author Response

Point 1: The number of hospitals and interviewees (27 key stakeholders, Number of participants) involved in the analysis is too small. 

Response 1: Thank you for your suggestions. The sample size of this study was 27, which was not a particularly small one among qualitative studies(Vasileiou, K., Barnett, J., Thorpe, S., & Young, T. Characterising and justifying sample size sufficiency in interview-based studies: systematic analysis of qualitative health research over a 15-year period. BMC medical research methodology, 2018, 18, 148.). An appropriate sample size for a qualitative study is one that answers the research question(Marshall, M. N. Sampling for qualitative research. Family practice. 1996, 13, 522-526.).In general, fewer participants are needed if the research question is narrower and participants are more diverse(Malterud, K., Siersma, V. D., & Guassora, A. D. Sample size in qualitative interview studies: guided by information power. Qualitative health research. 2016, 26, 1753-1760.). In this study, the research question was how public hospitals, private hospitals, and mixed-ownership hospitals in Guizhou Province differed in their hospital governance structure and organizational behavior. To increase the information power (the information the sample holds)( Malterud, K., Siersma, V. D., & Guassora, A. D. Sample size in qualitative interview studies: guided by information power. Qualitative health research. 2016, 26, 1753-1760.), we purposively sampled our informants to make our sample typical and representative. We sampled hospital managers from public hospitals, private hospitals, and mixed-ownership hospitals. We oversampled mixed-ownership hospitals because these hospitals were less understood compared to public and private hospitals. To ensure the diversity of hospital samples with mixed-ownership, we sampled both level II and level III hospitals as well as both general hospitals and specialized hospitals. We sampled 3 mixed-ownership hospitals as there were only a small number of mixed-ownership hospitals. Also, we sampled policymakers from the Ministry of Health and experts in nationwide hospital reform to supplement information collected in Guizhou Province. Lastly, we acknowledged that our findings might not generalize to all other regions of China (see the section of “research limitations”), just as other case studies.

Point 2: English sentences are not smooth. For example, Line 45 to 48, English word duplication is awkward. (Level I provides.. level III provides.., public hospitals and private hospitals, public hospitals can)

 Response 2: Thanks for your kind suggestion. We have checked the manuscript again to smooth the language.